# Referral Compliance Following a Diabetes Screening in a Dental Setting: A Scoping Review

**DOI:** 10.3390/healthcare10102020

**Published:** 2022-10-13

**Authors:** Andre Priede, Phyllis Lau, Ivan Darby, Mike Morgan, Rodrigo Mariño

**Affiliations:** 1Melbourne Dental School, University of Melbourne, Parkville 3010, Australia; 2Faculty of Dentistry, University of Otago, Dunedin 9054, New Zealand

**Keywords:** type 2 diabetes, prediabetes, referral compliance, oral health professionals, screening

## Abstract

With type 2 diabetes prevalence increasing in Australia, and the condition associated with significant morbidity and mortality, screening for dysglycaemia in the dental setting has been proposed to identify asymptomatic individuals. Screening commences with a risk assessment, and individuals identified at elevated risk for having diabetes are then referred to their medical practitioner for confirmation of their glycemic status. Therefore, for screening to be effective, individuals need to adhere to their oral health professionals’ (OHP) advice and attend their medical follow-ups. This review aims to investigate the literature on referral compliance following a risk assessment in the dental setting and identify barriers and facilitators to screened individuals’ referral compliance. A scoping review of the literature was undertaken, selecting studies of diabetes screening in a dental setting that recorded compliance to referral to follow-up, and explored any barriers and facilitators to adherence. Fourteen studies were selected. The referral compliance varied from 25 % to 90%. Six studies reported barriers and facilitators to attending medical follow-ups. Barriers identified included accessibility, cost, knowledge of the condition, and OHP characteristics.

## 1. Introduction

Globally, the International Diabetes Federation (IDF) estimates that 451 million adults have diabetes worldwide in 2017 and a further 352 million adults have impaired glucose tolerance (prediabetes) [1]. It is estimated that 1.2 million (6%) Australian adults aged 18 years and over had diabetes (type 1 diabetes and type 2 diabetes) in 2017–2018 [2]. In addition, the prevalence of previously undiagnosed diabetes may be approximately 25% of that of known diabetes [3] and one in six Australian adults have prediabetes [4].

Diabetes is associated with many acute and chronic complications. The chronic complications of diabetes include macrovascular complications (cardiovascular disease, stroke, peripheral vascular disease) and microvascular complications (retinopathy, nephropathy and neuropathy) [5]. The morbidity and mortality associated with the complications of diabetes are major contributors to healthcare costs, with the annual costs attributed to diabetes in Australia in 2010 estimated at $14.6 billion [6].

Amongst the chronic complications of diabetes is an increased prevalence, extent and severity of periodontal diseases (periodontitis and gingivitis) [7]. The relationship between diabetes and periodontal disease is bidirectional, as severe periodontitis may worsen glycemic control in people with diabetes [8]. Studies have shown that periodontal therapy may improve metabolic control and thus improve overall health outcomes for people with diabetes [9].

Prediabetes is characterised by elevated blood glucose, but not satisfying the diagnostic criteria for diabetes [10]. People with prediabetes are at high risk of developing type 2 diabetes and have an increased risk of cardiovascular disease, coronary heart disease, stroke, and all-cause mortality [11]. Women with prediabetes before pregnancy have a higher risk of developing Gestational Diabetes Mellitus [10]. Recent studies have also demonstrated that periodontal inflammation parameters including alveolar bone loss [12] and bleeding on periodontal probing [13], are worse in individuals with prediabetes compared to healthy individuals.

It has been estimated that three-quarters of people with diabetes receive a “delayed diagnosis,” meaning that within six months of diagnosis, they have already developed at least one diabetes-related comorbidity or complication [14]. Early detection of type 2 diabetes and prediabetes should enable optimal management, and in the case of prediabetes, reduce the risk of developing type 2 diabetes and cardiovascular disease. Screening can be defined as ‘‘the examination of asymptomatic people in order to classify them as likely or unlikely to have a disease” [15]. For individuals with undiagnosed prediabetes and type 2 diabetes, screening often represents the first step in obtaining a diagnosis of diabetes, which then may result in the initiation of interventions that can delay or even prevent diabetes-associated complications.

In Australia, the recommended protocol for screening for prediabetes and type 2 diabetes is administering a non-invasive risk assessment and referring individuals at increased risk of dysglycaemia to a General Medical Professional (GP) for diagnostic follow-up and management [16]. The Australian Type 2 Diabetes Risk Assessment Tool (AUSDRISK) was developed for the Australian population and offers a simple, non-invasive assessment to detect adults at elevated risk of dysglycaemia [17,18]. Those individuals identified as being at high risk are recommended to receive either a fasting venous blood glucose test, HbA1c or Oral Glucose Tolerance Test [16].

Most disease screening in Australia occurs opportunistically, often undertaken when patients attend healthcare professionals for reasons unrelated to the disease. When individuals consult with a GP, it is often for a problem and not a checkup [19], and awareness and application of the AUSDRISK has been found to be low amongst Australian GPs [17]. Given the bidirectional relationship between diabetes and periodontal disease, the oral manifestations of diabetes are common and readily identifiable by oral health professionals (OHPs). Since a significant proportion of the population has an annual dental check-up [20], the dental setting affords an additional location for administering a diabetes risk assessment.

There has been research exploring the feasibility and acceptability of opportunistic diabetes screening in the dental setting, but less is known about the referral compliance of individuals who were recommended to attend medical follow-up with a GP following a diabetes risk assessment by their dentists.

Therefore, the aims of this scoping review are to identify:the rates of referral compliance following diabetes screening in the dental settingthe barriers and facilitators to patients’ referral compliance following diabetes screening in the dental settingbehavioural models that have been developed to explain patient compliance to referral recommendations from OHPs following diabetes screening.

## 2. Methodology

The key search terms and their Medical Subject Headings (MeSH) terms and synonyms used in this scoping review were combined using Boolean operators:dysglycaemia or dysglyc* or type 2 diabetes or type two diabetes or diabetes mellitus type 2 or diabetes mellitus type two or type II diabetes or prediabetes or pre-diabetes or prediabetic or prediabetic state

And

screen* or detect* or test or testing or diagnos* or assess*

And

oral or dental or dentist* or oral hygienist* or dental hygienist* of oral health therapist* or dental therapist*

And

refer* or follow-up

And

comply or adhere* or attend*

We searched two online biomedical research databases: MEDLINE(OVID) and CINAHL. The search was limited to research published in the last 20 years, English language articles, research involving humans and articles that included an abstract. The exclusion criteria included screening for diseases/conditions other than diabetes and in locations that were not in a dental setting, studies on referral compliance of children and adolescents, and studies that explored only provider perspectives on diabetes screening. Papers were screened for relevance by one reviewer (AP) for title and keywords and the bibliographies of the final retrieved articles were screened for additional relevant literature.

## 3. Results

The original search yielded 774 potentially relevant articles. After duplicates were removed and abstracts and titles were screened for relevance, 71 publications met the eligibility criteria, and the corresponding full-text papers were obtained for review. Papers were excluded after full-text assessment for the reasons provided in the flowchart (Figure 1).

After screening the full-text articles, fourteen publications [21,22,23,24,25,26,27,28,29,30,31,32,33,34] were found that reported referral compliance for individuals discovered to be at an elevated risk of dysglycaemia in the dental setting (Table 1). These studies were conducted in Australia, the UK (two studies), the USA (seven studies), Sweden, Spain, Saudi Arabia and Germany. Researchers followed up on referral compliance from two weeks [21] to three years [22] following the diabetes risk assessment and referral by the OHP, to determine follow-up attendance rates.

### 3.1. Rates of Referral Compliance following Diabetes Screening in the Dental Setting

The reported referral compliance ranged from 25 % in the Australian study [32] to 90% in the Swedish publication [22]. In the Spanish study, the glycemic status of 14% of screened participants remained unknown and referral non-compliance may have contributed to this [33].

Two studies, in the UK and Australia, stratified referral compliance to the individuals’ diabetes risk assessment result, categorising patients as low, medium, and high risk of having dysglycaemia. Wright et al. found that 31% of patients identified as moderate-risk, and 20% of high-risk patients, visited their GP for follow-up [25]. Mariño et al. reported that 25 % of individuals identified as high risk of having diabetes and 21 % in the intermediate risk group, attended medical follow-ups [32].

### 3.2. Barriers and Facilitators to Patients’ Referral Compliance following Diabetes Screening in the Dental Setting

The reasons underlying non-compliance to follow-up recommendations were recorded in six papers (Table 2), and these included cost, misplacing the referral letter, being too busy to attend, moving overseas, delaying the appointment until after Ramadan, fear and distress about the results, perceiving the screening test to be a burden, believing the condition was not ‘serious enough’ to visit their GP [25] and other health conditions taking priority [32]. In the Australian study conducted in 2020, two participants cited concerns about COVID-19 as a reason for not complying with referral advice [32].

Where facilitators to attending the GP for follow-up (Table 3) were reported, these included the individual perceiving a positive screening result as an opportunity to act, experiencing a positive interaction with the healthcare professional (HCP), appreciating the ease with which results could be shared with their GP, motivation from observing family members experiencing diabetes or the desire to act as a role model for other family members [28]. Herman et al. identified three patient characteristics: having a history of tooth loss or dyslipidemia and older age increased the likelihood of referral compliance [29]. In this study, participants aged 30 years or older, were screened using random capillary glucose and a periodontal examination. Individuals with either elevated random capillary glucose and/or periodontitis that attended medical follow-up were on average 57 years of age compared with 54 years old for those who did not attend. For those participants with normal random capillary glucose and no periodontitis, those that complied with referral advice were on average 54 years old, compared to 50 years of age for those who did not attend.

In one study, a significant association was found between the number of ‘at risk’ screening results received and whether or not the individual would follow the advice and contact a GP. Patients were three times more likely to contact their GP if they received a positive risk result on both screening tools [24].

The dental settings where the diabetes screening was conducted, varied from private dental practices to public/community, and University healthcare locations. In the USA, Genco et. al. found that 79 % of patients with elevated HbA1c detected in a public community dental clinic attended their GP following referral, compared to only 21.5 % of individuals referred from private dental clinics [30]. The authors suggested this disparity may be due to the different demographic characteristics of the screening cohorts. The majority of individuals screened in the public clinic were African American or Hispanic, coming from communities with high levels of existing diabetes, and therefore, they may have had an increased awareness of the condition and be more willing to attend follow-up. Additionally, the medical and dental clinics were located within the community centre and they shared electronic records, making it easier for participants from these locations to attend follow-up.

### 3.3. Behavioural Models Developed to Explain Patient’s Compliance to Referral Recommendations from OHPs following Diabetes Screening

Only one publication, by Rosedale et al., presented a model to explain an individuals’ referral compliance behaviour following a diabetes risk assessment in the oral healthcare setting [28]. The authors adapted a model developed to explain barriers to hypertension treatment and follow-up [35], that identified four factors: intention, capability, healthcare system and fear and denial barriers that influenced referral compliance.

## 4. Discussion

This scoping literature review provides a picture of referral compliance following diabetes screening in the dental setting and the barriers and facilitators to compliance. Only three papers [27,28,30] reported exploring the factors that influence referral compliance amongst their primary objectives, with only one paper [28] adopting a thematic analysis of interviews of screened patients to specifically explore why they sought the recommended follow-up or not.

Screening for type 2 diabetes and prediabetes in the dental setting represents a continuum that begins with a risk assessment and concludes with individuals at an elevated risk of dysglycaemia, being referred to a medical professional (GP) for a definitive diagnosis. Thus, completion of the screening pathway requires individuals who are referred to their GPs to attend medical follow-ups, otherwise, their glycaemic status remains unknown. To evaluate the effectiveness of diabetes screening in the dental setting, it is important to know how many individuals comply with OHPs’ referral recommendations, and the barriers and facilitators to their compliance. By understanding what influences referral compliance, interventions may be developed to improve the rates of medical follow-up, thus maximising the effectiveness of diabetes screening.

### 4.1. Rates of Referral Compliance

For the papers included in this review, the mean referral compliance was 54% of referred patients. The highest referral compliance rate of 90% was reported in the Swedish study [22] while the lowest rate of 25% was reported in the Australian study [32]. It should be noted that some studies followed up with participants for up to three years after screening [22] while others followed up for only two weeks to six months [24,25,26]. Furthermore, confidence intervals were not always reported for all referral compliance rates, making it difficult to ascertain the variations or make comparisons. Nevertheless, the generally low level of referral compliance amongst screened patients represents a considerable barrier to effective diabetes screening in the dental setting. Comparable low rates of referral compliance have been reported in the literature for opportunistic diabetes screening in other primary care settings, including pharmacies and emergency departments. A systematic review and meta-analysis of diabetes screening conducted in community pharmacies concluded that although screening identified a significant number of individuals at a high risk of having diabetes, many of these individuals did not attend a follow-up appointment with their HCP [36]. Studies reporting referral compliance after diabetes screening in an Emergency Department have found attendance rates for a confirmatory follow-up to be between 38% [37] and 73% [38].

Similarly, research investigating referral compliance following screening for other medical conditions has found low rates of follow-up attendance. When the dental setting has been utilised to screen for sleep apnoea, less than half of referred participants attended medical follow-ups [39,40]. Screening efforts for diabetic eye disease [41], elevated cholesterol [42], and follow-up after an abnormal pap smear result [43] have also been hampered by low rates of compliance to post-screening referral.

### 4.2. Barriers and Facilitators to Referral Compliance in the Dental Setting (Patient Factors)

Factors impacting compliance to referral are complex and are linked with parameters, such as age, gender, stage of condition and disease complications [41,43].

Six publications reported patients’ reasons for choosing to comply with referral advice or not, after diabetes screening in a dental setting. One study identified patient demographic characteristics that may predict attendance to medical follow-up, finding older participants were more likely to comply with referral advice than younger participants although no specific reasons were elucidated [29]. Research in other primary care settings has reported demographic variables, such as increasing age for cervical cancer screening [43], have facilitated referral compliance following the medical screening. Hui et al. suggested this may be because younger women (under 30 years old) might be less able to manage the psychological distress associated with cervical cancer risk and its medical follow-up. Social and economic factors, such as unemployment, lack of social support, andlow education level, have also been found to negatively influence attendance following screening for cervical cancer [44].

The diabetes risk assessment results are interpreted by the OHP, who then communicates the findings to the participant. If a referral is indicated, it is the choice of the individual to act upon the advice or not. The way risk is perceived by the screening participant may influence whether they perform behaviours that reduce or prevent the risk [45]. The perception that diabetes was not serious enough to justify a follow-up visit [25,28], or that other health concerns took precedence [32], were reported as reasons for non-compliance in several studies in this review. Rosedale et al. found that patients would either perceive diabetes screening as being an opportunity to act, and this would encourage referral compliance, or a burden, and therefore, not attend medical follow-up [28].

An individual’s knowledge about the medical condition in question may also influence their perception of risk [45]. A lack of knowledge about diabetes [28] was reported as a barrier to referral compliance in this review. Knowledge about the condition being screened for has been reported as a facilitator of referral compliance for some conditions, such as type 2 diabetes [46] and a barrier, because of the distress and fear knowledge of the condition creates for breast [47] and cervical cancer [44].

The screening protocol employed in the dental setting may also influence the participants’ risk perception and thus referral compliance. A study that employed two screening tools: A questionnaire and a point of care Hb A1c test, found the number of positive screening results a participant received would influence their decision whether to attend a follow-up or not. Individuals that screened positive for both tests, were three times more likely to attend a medical assessment [24]. A study that explored patients’ perceptions of, and reactions to a diabetes screening protocol in the primary care setting, found patient perceptions changed as they progressed through a three-step screening pathway [46]. The initial screening steps were often viewed as unimportant, and it was not until the final diagnostic test (step three), that type 2 diabetes was considered a strong possibility. The authors suggested that participants experienced a process of psychological adjustment from the first screening test which was approached without considering its implications, to the final test where they were confronted with the possibility of having diabetes.

Diabetes screening conducted in the dental setting requires a referral pathway from OHP to GP, and one potential barrier to completing the screening protocol identified by Mariño and colleagues [32] was arranging a medical follow-up appointment for screened individuals. Less than half of screened respondents in this study reported that the booking of medical appointments was easy, and this may have contributed to some individuals at high risk of developing diabetes, delaying or not receiving a follow-up assessment from their GP. However, part of this research was conducted during the COVID-19 pandemic when restrictions on movement were in place, and two study participants cited COVID-19 as a reason for not attending follow-up. Additional barriers reported in the literature that may hinder booking a medical follow-up appointment have included cost [25,32], the individual relocating overseas [25,32], time constraints [21,25], cultural and religious reasons [25], and in one study simply losing the referral letter [25].

### 4.3. Barriers and Facilitators to Referral Compliance in the Dental Setting (OHP Factors)

Several papers in this review identified OHP factors, such as their communication skills, and the OHP and patient relationship, as a predictor of patients’ compliance with referrals. Rosedale et al. reported having a positive interaction with the OHP and good communication skills increased the likelihood of referral compliance [28]. Studies of disease screening have identified the role of HCP factors, such as effective communication [39], the screened individual having an existing relationship with an HCP [48], and positive interactions between the HCP and patient [46], facilitating referral compliance. Conversely, poor communication and a lack of rapport with the HCP has been identified as a barrier to referral adherence [49]. This suggests that the OHP providing information about the medical condition being screened for, engaging participants in the discussion, and as reported in one study, organising post-screening reminder calls [28], may encourage referral compliance.

The location of the oral health service where screening was undertaken, influenced the referral compliance in one study which found 78 % of individuals screened in a community clinic attended medical follow-up, compared with 21 % of those that were screened in a private dental practice [30]. The underlying reasons for this difference are not clear, as most participants simply chose not to seek care without providing an explicit explanation.

### 4.4. Behavioural Model That Explains Patient’s Compliance to Referral Recommendations

This scoping review identified only one paper that proposed a behavioural framework that may explain an individual’s decision to attend medical follow-ups post-diabetes screening [28]. Adapting a model developed to explain barriers to hypertension treatment and follow-up [35], Rosedale et al. proposed that intention, capability, healthcare system, and fear and denial barriers, influenced referral compliance. They categorised intentions, as the attitudes and motivation towards seeking medical follow-up. Capability barriers related to knowledge about the condition (type 2 diabetes) and understanding the importance of follow-up. Healthcare system barriers included affordability, acceptability and access to medical care. The final component of the model was fear of the consequences of positive screening, participants’ denial about the association between prediabetes and diabetes, and not believing themselves to have diabetes [28]. The development of a behavioural model enables us to understand behaviours and the factors that influence them. It is only by understanding the determinants of behaviour, that effective evidence-based interventions may be designed to change it.

### 4.5. Strengths and Limitations of This Review

The strength of this scoping review is that it focuses on a critical step in the diabetes screening pathway in the dental setting: compliance with the OHP referral for medical follow-up. It contributes to knowledge regarding the broad topic of compliance to referrals following a diabetes risk assessment in the dental setting and identified gaps in the existing literature. A limitation of this scoping review is that some relevant studies may have been overlooked due to the exclusion of studies published in a language other than English, e.g., other than one paper from Saudi Arabia [31], no developing countries are represented in this review.

## 5. Conclusions

This scoping review revealed literature on referral compliance following diabetes screening in the dental setting. In the studies included, there was a low rate of referral compliance amongst individuals screened for dysglycaemia in the dental setting. This low level of referral compliance was found in other studies exploring adherence to follow-up recommendations following screening for diabetes and other medical conditions in primary care settings. This review found limited studies exploring the reasons underlying the low referral compliance rates. It appears from current research, that non-compliance to medical follow-up following a diabetes risk assessment is influenced by multiple determinants, including demographic, patient knowledge attitudes and beliefs regarding the condition, HCP provider factors, such as the HCP–patient relationship, the rapport between HCP and communication and healthcare system factors.

The findings from this scoping review highlight a gap in our knowledge regarding the barriers and facilitators that influence referral compliance following a diabetes risk assessment in the dental setting. Therefore, further research is needed to understand what the barriers and facilitators to referral compliance are following a diabetes screening in an Australian dental setting. By being able to understand patient and provider characteristics that influence adherence to referral, a behavioural model may be developed that allows for interventions to be designed to maximise the number of patients that follow up the advice of their HCP and complete the diabetes screening pathway.

## Figures and Tables

**Figure 1 healthcare-10-02020-f001:**
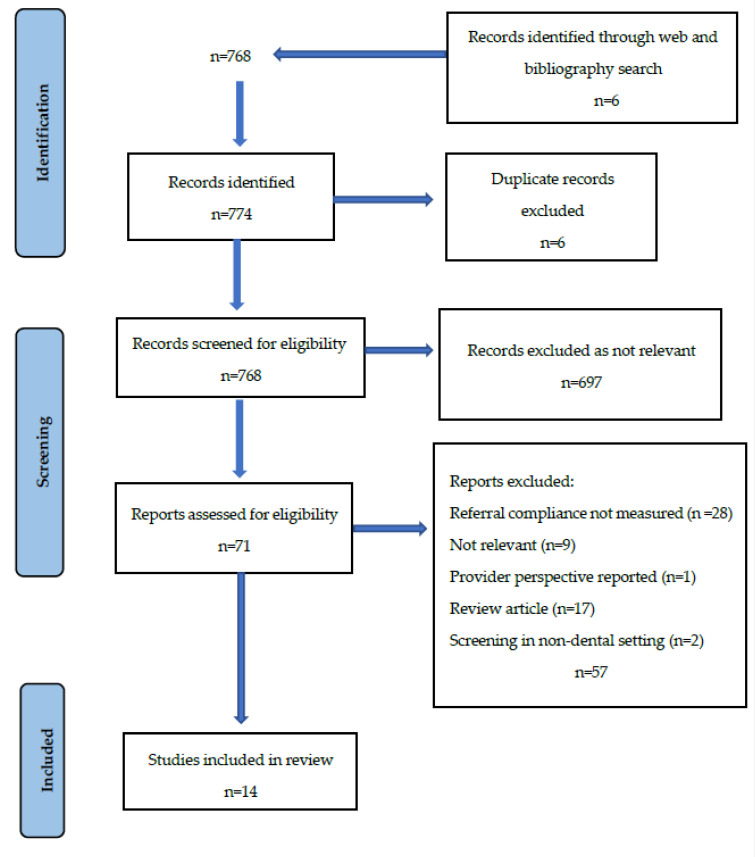
PRISMA flowchart of study selection process.

**Table 1 healthcare-10-02020-t001:** Scoping review results.

Main Author Date	Objectives	Country	Screening Instruments Used	% Referral Compliance	Barriers and Facilitators to Referral Compliance	Results and Conclusions
Bossart et al., 2015 [21]	Assess the effectiveness, convenience and cost of POC diabetes screenings performed by a dental hygienist for patients with periodontitis.	USA	Periodontal exam, risk assessment and POC-Hb A1c	53%	Waiting for next GP visit and time constraints	34% (n = 17) participants screened positive for dysglycaemia. Diabetes screening by dental hygienists is effective and convenient for identifying dyglycaemia.
Engstrom et al., 2013 [22]	Test the effectiveness of diabetes screening in a collaborative framework between oral and primary health care.	Sweden	Risk assessment using BMI, RCBG, and RPG.	90%	Not reported	9 individuals were diagnosed with T2D. Diabetes screening was successful in terms of response rate and referral compliance.
Franck et al., 2014 [23]	To investigate the ability to screen dental patients for T2D or PD.	USA	Risk assessment Survey. High-risk received POC and labor Hb A1c	55%	Not reported	28 participants had prediabetes. identified patients with dysglycaemia.
Bould et al., 2016 [24]	To determine dental patients’ uptake of two screening tools FR and POC-HbA1c. in general dental practice.	UK	FR screening tool and POC-HbA1c	60%	Significant association between number of ‘at risk’ screening results received and whether or not patient attended follow-up.	258 participants identified as at risk of diabetes. A two-step method of diabetes screening was acceptable to patients, and the majority complied with referral advice. Patients were three times more likely to contact their GP if they received a positive risk result on both screening tools.
Wright et al., 2013 [25]	To assess the feasibility of implementing a T2D diabetes risk assessment screening in dental settings using the NICE guidance tool	UK	Risk assessment	26% (30.6% moderate-risk and 20% of high-risk patients attended follow-up)	Barriers to medical follow-up: misplacing the referral letter, being too busy, being away, delaying the appointment until after Ramadan and perception condition was not ‘serious enough’ to visit their GP.	Diabetes screening is feasible in dental settings. Amongst the challenges to this approach for the OHPs are time constraints, limited manpower and low referral compliance
Ziebolz et al., 2019 [26]	Investigate the efficacy of T2D screening based on questionnaire replies	Germany	FR-positive patients were referred to a specialist	55%	Not reported	The survey tool identified patients with T2D and prediabetes and is suitable for diabetes screening in dental practices. Difficult to motivate individuals to attend medical follow-up
Biethman et al., [27]	To evaluate GPs’ responses to requests for information regarding follow-up results after diabetes screening in a dental setting. A secondary aim was to evaluate patients’ referral compliance.	USA	POC Hb A 1C	59%	Unable to predict patients’ compliance with seeking follow-up with their GPs.	Most patients complied with their OHP’s advice to seek medical follow-up after diabetes screening. A single written request from an OHP to the GP to share the results may be insufficient and a phone call may be a more effective communication method.
Rosedale et al., 2017 [28]	To examine patient experiences after receiving positive diabetes screening results at a dental clinic, whether they attended medical follow-up and facilitators and barriers to referral compliance.	USA	Hb A1c	54 %	Facilitators: positive screening result viewed as an opportunity to act, 3-month follow-up/ reminder call from OHP. Barriers: Positive screening results perceived as a burden, lack of knowledge about diabetes, not understanding the importance of follow-up, busyness, financial concerns, fear and denial.	Patients and OHPs believe the dental setting is an acceptable and feasible site for diabetes screening. A limitation of diabetes screening is the extent to which patients’ follow-up positive screening results with their GPs.
Herman et al., 2015 [29]	To develop and validate a tool to screen for PD and T2D in dental practices	USA	Risk assessment survey, RCBG and periodontal exam	26%	Those that complied were significantly older than those who did not. More likely to comply if a previous history of tooth loss or dyslipidaemia	30 % of patients ≥30 years old seen in general dental practices had dysglycaemia. Screening for dysglycaemia can be used to identify high-risk patients.
Genco et al., 2014 [30]	To assess patient compliance with referral to GPs for diabetes diagnosis	USA	Risk assessment and POC-HbA1c	35%	78.8 % of patients from community clinics and 21.5 % were referred from private dental clinics attended medical follow-ups. Patients reported they declined to seek follow-up without giving an explicit reason.	Patients and OHPs support diabetes screening in the dental setting. Low referral compliance occurred in the private dental setting and good compliance in the community health centre setting. The reasons for low referral compliance need to be investigated and addressed before screening for diabetes in the dental setting can be advocated.
Al Ghamadi et al., 2013 [31]	To assess the efficacy of the dental setting for T2D and PD screening	Saudi Arabia	Random blood glucose levels (RBGLs) were recorded.	84%	Not reported	16.4 % undiagnosed T2D and 15.8 % PD among patients visiting dental clinics
Marino et al., 2020 [32]	To develop and evaluate an innovative approach for identifying pre-diabetes and type 2 diabetes within the private oral health setting.	Australia	AUSDRISK risk assessment tool	25%	Cost, personal issues, other health concerns taking priority and COVID-19, were named as barriers to attending medical follow-ups.	Six individuals were diagnosed with prediabetes. T2D screening in a dental setting is well-accepted and effective. However, developing referral pathways, both to and from GPs, as well as maximising follow-ups is required.
Montero et al., 2020 [33]	To evaluate the efficacy of different screening protocols for undiagnosed dysglycaemia in the dental setting	Spain	FR screening tool and periodontal exam and POC-HbA1c	Results of 23 referred patients were unknown, for some, this may be due to referral non-compliance.	Not reported	8.5% of individuals were diagnosed with dysglycaemia. The screening protocol was feasible and effective in identifying participants with dysglycaemia in the dental setting.
Lalla et al., 2015 [34]	To assess an approach to improving behavioural and glycaemic outcomes in dental patients with diabetes risk factors and previously undiagnosed hyperglycaemia	USA	Risk assessment, periodontal exam, and Hb A1c	84%	Not reported	At 6 months most of the participants reported having attended a GP and 49% reported at least one positive lifestyle change. In participants identified as at risk of diabetes, HbA1c was significantly reduced.

Glossary of terms: T2D—type 2 diabetes, PD—prediabetes, AUSDRISK—Australian type 2 diabetes risk assessment tool, GP—medical professional, POC—point of care, HbA1c—glycosylated haemoglobin, BMI—body mass index, RCBG—random capillary blood glucose, RPG—random plasma glucose and FR-FINDRISC—Finnish Diabetes Risk Score.

**Table 2 healthcare-10-02020-t002:** Barriers to participants adhering to referral recommendation of the OHP.

Barriers to Referral Compliance	Study
Referral pathway issues (e.g., negative perception of services losing the referral, no longer in the country)	[21,25,32]
Too busy to attend	[21,25,28]
Cultural/religious reasons (e.g Ramadan)	[25]
Lack of knowledge and awareness of the condition	[25,28]
Fears and distress about the results	[28]
Cost of follow-up appointment	[28,32]
Other health issues took priority	[32]
Patient perceived positive screening result as a burden	[28]

**Table 3 healthcare-10-02020-t003:** Facilitators to participants’ adhering to referral recommendation of the OHP.

Facilitators to Referral Compliance	Study
Good HCP–patient interaction	[28]
Good interprofessional communication between dental and medical professional	[28]
Patient perceiving positive screening as an opportunity to act	[28]
Observing family members with diabetes or desire to be a role model for family members	[28]
History of tooth loss and dyslipidaemia	[28]
Location of screening; Community dental clinic	[30]
Receiving a reminder to follow-up	[28]
Multiple ‘at risk’ screening results received	[24]

## Data Availability

Not applicable.

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
