# Peer review of "Referral Compliance Following a Diabetes Screening in a Dental Setting: A Scoping Review"

_healthcare, 2022, doi:10.3390/healthcare10102020_

Round 1

Reviewer 1 Report

The review here is a findings from this scoping review and deserves consideration in this journal. The topic is within the aim and scope and the text is correctly organized. However I found some minor points to address before publication:
The author defines the purpose of the study as follows: the rates of referral compliance following diabetes screening in the dental setting, the barriers and facilitators to patients’ referral compliance following diabetes screening in the dental setting; and behavioural models that have been developed to explain patient’s compliance to referral recommendations from OHPs following diabetes screening. The results are presented but difficult for the reader to understand. Add the results of this analysis as a flow diagram.

Author Response

We thank Reviewer 1 for their encouraging comment that our manuscript deserves consideration by the journal.

We agree that the results as presented in a continuous format can be difficult for the reader to understand. We do not think that presenting the results as a flow diagram would make the results easier to understand since (a) Tables 1-3 already visually present the key results and (b) although the rates of referral compliance and barriers/facilitators may present well as a single flow chart, the behavioural model to explain patient’s compliance is unlikely to fit well.

Instead, we have added subheadings in the Results section, corresponding to the three purposes of the literature review, to make it easier for readers to read. We hope this addresses Reviewer 1’s concerns.

Reviewer 2 Report

This is a very useful and appropriate article.  Early detection  and/or treatment are essential for primary and secondary prevention.  this article capture the importance of that principle.  Diabetes is increasing globally for several reasons.  Perhaps more knowledge about the bi-directional risk factors associated with diabetes and periodontal disease will increase awareness for patients and providers across all healthcare disciplines.  This will also impact healthcare costs when diabetes can be controlled and the majority of patients living with diabetes have a dental home.  We must work closer across healthcare disciplines to reinforce best practices.

Note line #158 has a typo correction to be made.

Author Response

We thank Reviewer 2 for taking the time to provide feedback and your encouraging comments regarding the manuscript.

Thank you for alerting us to the typo and we have corrected the typo on line #158.

Reviewer 3 Report

The article is an extremely well written and structured review of a relevant topic of screening type 2 diabetes and its challenges for dental disorder correlation. Overall I really appreciate the structure, and the article is easy to read, furnished with relevant information. I have a few comments for the article.
1) Regarding Table1. - Can the authors incorporate a confidence index wrt the statistics involved in the 14 articles reported. This is a natural question that arises when for some countries like Sweden reports 90% compliance while others such as Australia reports about 25%. If there are no developing countries being reported in literature, I believe it also requires a mention in the discussion/limitations. Besides, the number of participants reporting this compliance has to be linked with parameters such as age, sex, stage of T2DM and/or not limited to simultaneous complications (which is mentioned in the introduction). Overall, I think the results and conclusions column can be much better represented.

2) Some of the parameters listed in Table 2 looks like behavioral such as Ref#28 or 32. And even sections 4.3 and 4.4 seems to be overlapping. Would the authors care to distinguish how OHP factors are different from the behavioral model for the distinctions - this recommendation may help understand a broader readership.

3) Kindly refresh the reference checks. A cursory verification revealed Ref#25 and 28 are not associated with Cultural/religious reasons as mentioned. 
Besides, for Table3, all parameters have been determined based on one reference #28. I do hold reservation for too much reliance on one article to come up with a conclusion that may be interpreted differently. If possible kindly structure the results section in a way the cited article is mentioned with the deserved weightage.

Author Response

We thank reviewer 3 for your valuable feedback and comments.

1.  With respect to comments about Table 1, unfortunately the statistics reported in the 14 selected papers did not include confidence intervals. We have now acknowledged this in Discussion by amending the second sentence in 4.2 – “The highest referral compliance rate of 90% was reported in the Swedish study (ref) while the lowest rate of 25% was reported in the Australian study (25). It should be noted that some studies followed up participants for up to three years after screening(22) while others followed up for only two weeks to six months (25,28). Furthermore, confidence intervals were not reported for referral compliance rates, making it difficult to ascertain the variations or make comparisons.”

We have specifically acknowledged the limited representation of developing countries in this review with an amendment to the last sentence in Discussion – “A limitation of this scoping review is that some relevant studies may have been overlooked due to the exclusion of studies published in a language other than English eg. other than one paper from Saudi Arabia (31), no developing countries are represented in this review.”

From the literature reviewed, it was clear that there are many complex and interacting parameters that impact referral compliance. We have now added this to the start of 4.2 – “Factors impacting compliance to referral are complex and are linked with parameters such as age, gender, stage of the diabetes and disease complications (41,42,43).”

2. The behaviour of interest in this literature review is ‘attending medical follow-up after a diabetes risk assessment’, and this is influenced by multiple factors. Table 2 reports the way these influences have been categorised in the papers found in this review. We agree that there is a degree of overlap between these influences, which reflects the way influences of behaviour may be connected and interact with each other.  My intention following on from this review is to use the COM-B behavioural framework to help understand referral compliance, and identify influences that may be targeted to enable an increase in referral compliance. However, the mention of COM-B is outside the scope of this literature review report.

3. Thank you for alerting us to the referencing error. We have deleted ref 28 from cultural/religious barriers from Table 2.

We have retained ref 25, as it reports that for some participants fasting for Ramadan was a barrier to attending medical follow-up.

4. We understand the reviewer’s concerns, although refs 24 and 30 were also cited in Table 3, not just ref 28.  This table relates only to facilitators of referral compliance and table 2(barriers) includes additional references . The majority of articles found for this scoping review did not attempt to explore the factors influencing referral compliance as their primary objectives. Only ref 28, amongst its primary objectives, explored whether positive-screened patients sought recommended follow-up, using thematic analysis of interviews with screened individuals. We have added a paragraph at the start of Discussion to provide this context – “This scoping literature review provides a picture of the referral compliance following diabetes screening in the dental setting and the barriers and facilitators to compliance. Only three papers (27,28,30), reported exploring the factors that influence referral compliance amongst their primary objectives, with only one paper (28) adopting a thematic analysis of interviews of screened patients to specifically explore why they sought recommended follow-up or not.”

In Conclusions, we have already highlighted  the need for more research to investigate the underlying factors that influence referral compliance behaviour.

Round 2

Reviewer 3 Report

Thank you for considering the reviewers comments for the article. I recommend publication of the article in its current form.